# Generative Modeling of Protein Conformational Ensembles With Cryo-EM Density Map Diffusion

**Jay Shenoy**
Department of Computer Science
Stanford University, Flatiron Institute, & SLAC National Lab
Stanford, CA 94305, USA
`jshenoy@stanford.edu`

**Axel Levy**
Department of Electrical Engineering
Stanford University & SLAC National Lab
Stanford, CA 94305, USA

**Miro Astore**
Center for Computational Biology
Flatiron Institute
New York, NY 10010, USA

**Frédéric Poitevin**
LCLS Data Systems
SLAC National Lab
Menlo Park, CA 94025, USA

**Sonya Hanson**
Center for Computational Biology
Flatiron Institute
New York, NY 10010, USA

**Gordon Wetzstein**
Department of Electrical Engineering
Stanford University
Stanford, CA 94305, USA

## Abstract

Knowledge of a protein's conformational ensemble is critical to determining its function, yet state-of-the-art ensemble prediction models are limited by lack of high-quality conformational data from simulation or experiment. Recent advances in heterogeneous reconstruction for cryo-electron microscopy (cryo-EM) have enabled scientists to visualize ensembles of conformational density maps for larger proteins and complexes not typically accessible through simulation, but building atomic models into these maps remains a challenge. In this work, we propose a diffusion model, CryoSampler, that learns to generate atomic conformational ensembles while only being trained on cryo-EM maps and static atomic reference structures. By framing the optimization objective as a map denoiser that internally consists of an atomic predictor followed by a differentiable forward model of the cryo-EM volume rendering process, we show that CryoSampler properly estimates atomic coordinates for the training set and generalizes to new proteins at inference time via unconditional sampling, without needing any density maps. We demonstrate these capabilities on a synthetic dataset of calmodulin proteins simulated with a bimodal distribution.

## 1 Introduction

Prediction of protein ensembles is widely regarded as the next frontier in computational structural biology [Lane (2023); Sala et al. (2023); Ourmazd et al. (2022)] following the success of AlphaFold [Jumper et al. (2021); Abramson et al. (2024)] in the modeling of static structures. The development of such dynamic models is hindered by the lack of high quality protein ensemble data available in public repositories. Today, X-ray crystallography accounts for the majority of the deposited structures in the Protein Data Bank (PDB), but cryo-electron microscopy (cryo-EM) has grown quickly in popularity in recent years [Berman et al. (2000)]. Notably, the imaging conditions of cryo-EM enable proteins to explore a larger space of conformations than is possible with crystallography. Cryo-EM is therefore a natural choice for training next-generation ensemble predictors as it also provides access to ensemble data that is difficult to probe via simulation due to molecular size constraints.

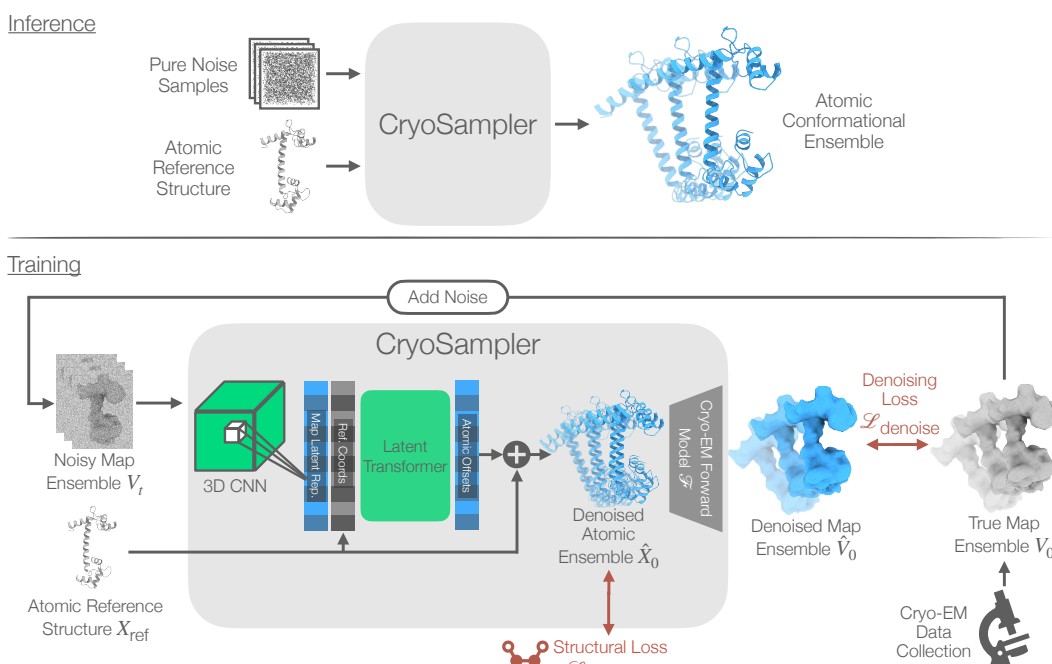

Figure 1: Illustration of inference and training pipelines. At inference time, CryoSampler can generate ensembles of protein structures by iteratively sampling from pure noise (and conditioning on a static reference). We train the model as a cryo-EM density map denoiser that operates by converting the noisy input maps to a set of estimated atomic coordinates, and then apply a differentiable cryo-EM forward model that transforms these coordinates into denoised maps. CryoSampler is supervised only on the set of observed maps $V_0$ and does not require atomic models corresponding to these maps. Unlike inference-time approaches like CryoBoltz [Raghu et al. (2025)] and ROCKET [Fadini et al. (2025)], our trained model can produce conformations for new proteins without requiring any density maps, solely a static reference structure.

Integrating cryo-EM data into model training is challenging because although algorithms can reconstruct ensembles of cryo-EM maps [Punjani & Fleet (2021); Zhong et al. (2021); Levy et al.], converting these maps into atomic structures, a process known as atomic model building, is complicated by the resolution of the heterogeneous volumes. Existing model building algorithms [Jamali et al. (2024)] often produce incomplete structures, but emerging methods [Fadini et al. (2025); Raghu et al. (2025)] have shown that models such as AlphaFold can serve as priors that enhance structure completion and accuracy. The use of generative priors in this setting ultimately begets the question: can atomic model building of cryo-EM maps be enveloped into the training of the structure prediction model itself, enabling accurate ensemble prediction?

This work establishes a generative model, CryoSampler, that learns to predict atomic protein ensembles supervised only on cryo-EM density maps, with model building performed implicitly in an end-to-end fashion. As initial validation, we demonstrate that our method can learn accurate ensembles from simulated cryo-EM maps of distinct calmodulin proteins that exhibit similar motion, highlighting its ability to jointly perform model building and generative ensemble prediction. Notably, at inference time, our method can predict conformations for proteins in the calmodulin family for which we lack cryo-EM maps, demonstrating in-domain generalization.

## 2    PRIOR WORK

Existing learning-based methods for ensemble prediction either operate at training time or inference time. In the case of the former, AlphaFold 3 [Abramson et al. (2024)] improves on the capabilities of AlphaFold 2 [Jumper et al. (2021)] by utilizing a diffusion model to produce atomic structures from latent pair and single representations of molecules. Due to its stochasticity, the diffusion model

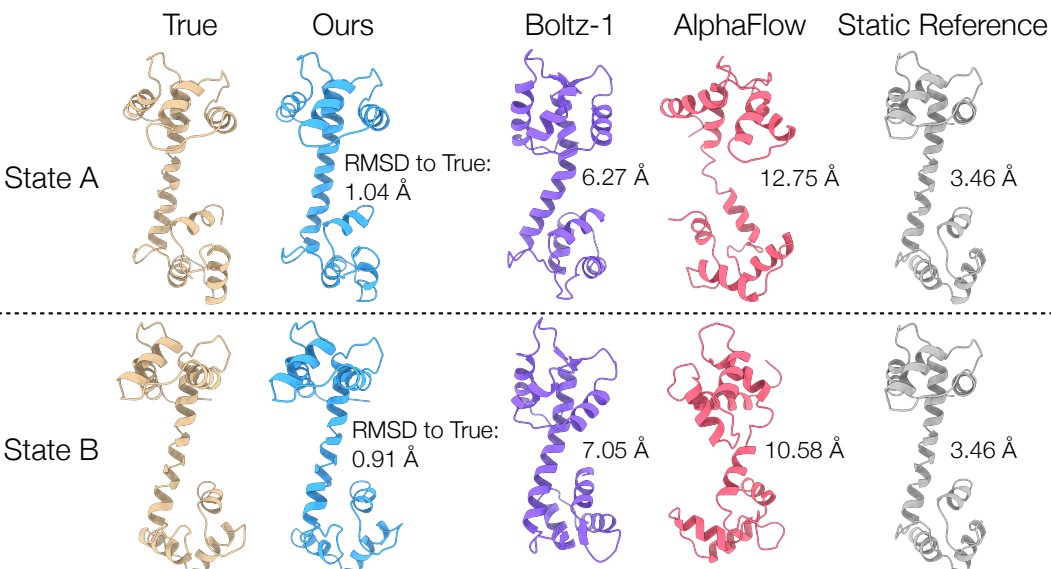

Figure 2: Evaluation on generating the simulated bimodal distribution of a test calmodulin protein (PDB: 4CLN [Taylor et al. (1991)]). This protein falls within our validation set and is therefore not seen during training. For each method, we generate 16 samples and determine the generated structure closest to each true state. Our method outperforms AlphaFlow [Jing et al. (2024)] and Boltz-1 [Wohlwend et al. (2025)] in terms of coverage, recovering the two states with near- or sub-angstrom RMSD. As AlphaFlow and Boltz-1 are not trained on states generated with normal mode analysis, we should not expect them to perform well on our input data, but we report their accuracy for the sake of completeness. We also report the distance of the reference structure from each state as a static baseline.

can technically produce ensembles of protein structures despite the explicit inclusion of large sets of ensemble data in its training set, but the interpretation of these outputs as true ensembles, as determined by comparison against ground truth data from experiment or simulation, is unproven as of yet. A contemporaneous work, BioEmu [Lewis et al. (2025)], extends the frozen trunk of AlphaFold 2 with a diffusion model trained on molecular dynamics (MD) data and experimental protein stabilities. Similarly, AlphaFlow [Jing et al. (2024)] performs ensemble prediction by repurposing AlphaFold 2 as a flow matching model trained on conformations in the PDB as well as MD data. While these techniques succeed in extending static prediction methods to the ensemble setting, they are inherently limited by the diversity of the training data, with MD facing practical limits on molecular size and simulation timescale, and the PDB lacking the full set of conformations one may observe in a cryo-EM experiment.

Inference time approaches, on the other hand, modify various components of pretrained models in order to generate alternative conformations. Wayment-Steele et al. (2024) and Monteiro da Silva et al. (2024) demonstrate that ablating the multiple sequence alignment (MSA) input of AlphaFold can lead to physically-accurate alternate structures confirmed by experiment. Recently, Suzuki & Amagasa (2026) have shown that manipulating the model's pair representation rather than the MSA achieves improved ensemble coverage. Similar to the training time approaches described previously, these inference time techniques are constrained by the relatively static nature of the training data from the PDB, as they only operate within the weight space of models trained on PDB (or self-distilled) data.

Our method is the first to propose training ensemble prediction models on heterogeneous cryo-EM reconstructions, circumventing the traditional two-stage model building and structure training pipeline with an end-to-end approach that supervises directly on cryo-EM maps. As a consequence, our technique has the potential to scale to larger systems that are challenging to simulate with MD, or that are difficult to reconstruct atomically from experimental cryo-EM data.

| | Training | | Validation | |
|---|---|---|---|---|
| | State A RMSD (↓) | State B RMSD (↓) | State A RMSD (↓) | State B RMSD (↓) |
| Ours | **0.29 Å** | **0.29 Å** | **1.10 Å** | **1.03 Å** |
| AlphaFlow | 8.42 Å | 6.18 Å | 9.88 Å | 6.49 Å |
| Boltz-1 | 6.64 Å | 6.94 Å | 6.38 Å | 7.00 Å |
| Static | | 3.46 Å | | |

Table 1: Results on synthetic calmodulin dataset. Our method overfits the training data to sub-angstrom accuracy, and generalizes well to the validation set with near-angstrom RMSD. We exceed the performance of AlphaFlow, Boltz-1, and a null "static" model that simply predicts the reference structure. AlphaFlow and Boltz-1 were not trained on normal mode analysis trajectories and are therefore not well-suited for modeling our dataset, but we include them here for completeness.

## 3 METHODS

CryoSampler is a generative model that is trained to denoise cryo-EM maps in volume space. Unlike prior methods that predict ensembles by training diffusion models on atomic structures, we supervise purely on the density maps that give rise to these structures, conducting model building implicitly during training. Following the framework set forth by Tewari et al. (2023) for novel view synthesis of real-world scenes, we perform diffusion in the space of observations (density maps), assuming that we lack access to the distribution of interest (that of atomic structures).

At a high level, our model learns the distribution of atomic offsets needed to deform a static atomic reference structure to produce an atomic ensemble that explains the training dataset of map ensembles, such that at inference time, one can sample atomic ensembles for proteins not seen during training. Internally, CryoSampler's architecture consists of a learned atomic encoder that converts noisy input maps into atomic coordinates, followed by a differentiable, physics-based forward model (decoder) that turns 3D atomic coordinates into cryo-EM density maps. Our model's pipeline is illustrated in figure 1 and is described in more detail in the appendix.

## 4 RESULTS

We test our model on a dataset of 6 calmodulin proteins retrieved from the PDB. Protein motion is simulated using normal mode analysis with the ProDy software package [Bakan et al. (2011)] along the first mode, from which the two extreme states are selected for training. We then simulate a cryo-EM map from each structure using the EMAN2 package [Tang et al. (2007)]. The structures are split into a training set with 4 proteins and a validation set with 2 proteins. After training is complete, we generate samples for the validation set and compute the RMSD of the closest generated structure for each state of every protein. As shown in figure 2 and table 1, our model recovers bimodal distributions for both validation proteins, outperforming Boltz-1, AlphaFlow, and a null reference prediction baseline in terms of all-atom RMSD.

## 5 CONCLUSIONS AND FUTURE WORK

In this work, we show that training a diffusion model directly on cryo-EM density maps end-to-end permits sampling of diverse atomic conformations at inference time. We initially validate our approach on a synthetic dataset of calmodulin proteins that exhibit similar conformational variety when simulated with normal mode analysis. The benefit of our method lies in its ability to train directly on the observables produced by cryo-EM, enabling it to access ensemble information that is inaccessible to crystallographic or simulation-based approaches. In the future, we hope to test our method on real cryo-EM maps from experimental datasets processed with heterogeneous reconstruction tools and to further improve its ability to generalize by conditioning on sequence-based features.

MEANINGFULNESS STATEMENT

We consider a "meaningful representation of life" one that treats raw observational data as the truth. In this work, these data take the form of cryo-EM density maps. Rather than modeling the atomic structures derived from these maps, we instead train a generative model on the maps directly and in doing so, implicitly infer the underlying atomic structures. We show for the first time that end-to-end diffusion on density maps with physics-based decoding permits joint modeling of experimental data and atomic models.

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

## A  APPENDIX

### A.1  TRAINING

We define the forward diffusion process using a continuous Variance Preserving (VP) SDE [Song et al. (2020b)] with a cosine schedule. The training data consists of ground truth cryo-EM density maps $V_0$ for various proteins, along with a static reference structure $X_{\text{ref}}$ assumed to represent the ground state of the ensemble (in practice, retrieved via PDB depositions or the outputs of pretrained models like Boltz [Passaro et al. (2025)]).

During training, the noisy maps input to the model are computed with a forward diffusion process that interpolates between the true maps $V_0$ and the standard normal distribution in volume space, $\varepsilon \sim \mathcal{N}(0, \mathbf{I})$, as follows:

$$V_t = \frac{\alpha_t}{\sigma_{\text{data}}} V_0 + \sigma_t \varepsilon, t \in [0, 1] \tag{1}$$

The variances are defined using a cosine schedule:

$$c(t) = \cos^2\left(\frac{t+s}{1+s} \cdot \frac{\pi}{2}\right), \tag{2}$$

$$\alpha_t = \sqrt{c(t)}, \sigma_t = \sqrt{1 - c(t)} \tag{3}$$

The optimization objective is to learn a set of weights $\theta$ that denoise the input maps $V_t$ via a learned atomic encoder $f_\theta$ that converts noisy maps to estimated atomic coordinates and a differentiable decoder $\mathcal{F}$ that represents the cryo-EM forward model between atomic coordinates and denoised density maps. Because the estimation of coordinates from density maps to atomic coordinates is an ill-posed inverse problem that becomes particularly pronounced as the maps degrade in resolution, we constrain the atomic coordinate estimation step with a set of structural losses that encourage the atomic encoder to produce atomic structures that are stereo-chemically valid. Thus, the objective function is a multi-part loss that balances between outputting denoised maps that match the input dataset as well as producing intermediate structures that are chemically sound:

$$\mathcal{L}_{\text{total}} = \mathcal{L}_{\text{denoise}} + \mathcal{L}_{\text{structural}} \tag{4}$$

The denoising loss is as follows:

$$\mathcal{L}_{\text{denoise}} = \mathbb{E}_{t, V_0, \varepsilon}\left[\min(\text{SNR}_t, \gamma) \cdot \frac{1}{\sigma_{\text{data}}^2} \left\|\hat{V}_0(V_t, \text{SNR}_t, X_{\text{ref}}) - V_0\right\|^2\right] \tag{5}$$

Here, $\hat{V}_0$ is the volume denoiser that produces intermediate denoised atomic coordinates $\hat{X}_0 = f_\theta(V_t, \text{SNR}_t, X_{ref})$, followed by $\mathcal{F}$, the forward model for cryo-EM. By composing these operations, the denoised volume becomes $\hat{V}_0 = \mathcal{F}(\hat{X}_0)$. We use a min-SNR loss weighting [Hang et al. (2023)] with $\gamma = 10$ to ensure proper weighting of different noise levels.

The chemical loss consists of two parts, one that regularizes the arrangement of all the atoms within each residue and another to constrain the local geometry of the backbone atoms:

$$\mathcal{L}_{\text{structural}} = \lambda_R \mathcal{L}_{\text{residue}} + \lambda_B \mathcal{L}_{\text{backbone}} \tag{6}$$

Both of these losses compare the local geometry of the predicted atomic structure with that of the reference, imposing soft rigid body constraints between the two. For the residue-level loss, we compute a term penalizing deviations from the inter-atomic distances of the reference structure:

$$\mathcal{L}_{\text{residue}} = \frac{1}{\sum_k |P_k|} \sum_{k=1}^{N_{\text{res}}} \sum_{(i,j) \in P_k} \left( ||\hat{X}_i - \hat{X}_j||_2 - ||X_{\text{ref},i} - X_{\text{ref},j}||_2 \right)^2 \tag{7}$$

Here, $P_k = \{(i,j) \in R_k \times R_k : i < j\}$ and $|P_k|$ is the number of valid pairs in the residue. Across residues, we compute the distances between the backbone atoms in a sliding window of $W = 5$ consecutive residues, again comparing this metric between the predicted and reference structures. The purpose of the backbone loss is to regularize the geometry of atoms across residues, such that features like alpha helices in the reference structure are preserved. The backbone loss is:

$$\mathcal{L}_{\text{backbone}} = \frac{1}{M \cdot |P_B|} \sum_{m=1}^{M} \sum_{(i,j) \in P_B} \left( ||\hat{X}_i - \hat{X}_j||_2 - ||X_{\text{ref},i} - X_{\text{ref},j}||_2 \right)^2 \tag{8}$$

Here, $M = N_{\text{res}} - W + 1$ is the number of sliding windows, $P_B$ is the pairwise combinations of the 20 backbone atoms within the window, $|P_B|$ is the normalization factor.

## A.2 SAMPLING

Sampling occurs over $K = 100$ steps, linearly discretized in the range $t \in [0, 1]$ such that $1 = t_1 > t_2 > ... > t_K = 0$. For each step $k$, the signal and noise scales $\alpha_{t_k}, \sigma_{t_k}$ are computed using the same continuous cosine schedule as in training. At each discrete step $k$, the model takes as input the current noisy volume $V_{t_k}$ and outputs a normalized denoised volume $\hat{v}_0$:

$$\hat{v}_0 = \frac{1}{\sigma_{data}} \hat{V}_0(f_\theta(V_{t_k}, \text{SNR}_{t_k}, X_{ref})) \tag{9}$$

The implied noise is then:

$$\hat{\epsilon}_k = \frac{V_{t_k} - \alpha_{t_k} \hat{v}_0}{\sigma_{t_k}} \tag{10}$$

The next volume is computed using the DDIM update rule [Song et al. (2020a)]:

$$V_{t_{k+1}} = \alpha_{t_{k+1}} \hat{v}_0 + \sigma_{t_{k+1}} \hat{\epsilon}_k \tag{11}$$

The volume used at the first step is $V_{t_1} \sim \mathcal{N}(0, \mathbf{I})$, which contains pure Gaussian noise. Following the final update at step $K$, the volume $V_{t_K}$ represents the fully denoised density at $t = 0$. To retrieve the definitive atomic conformational structure $X_{\text{final}}$, the volume is passed through the network $f_\theta$ with a noise conditioning signal corresponding to a noise-free state (SNR $\to \infty$, or $t = 0$):

$$X_{\text{final}} = f_\theta(V_{t_K}, \text{SNR}_{t \to 0}, X_{\text{ref}}) \tag{12}$$

This step allows the model to extract the 3D atomic coordinates from the denoised map, providing the physical coordinates of the protein conformation.

## A.3 ATOMIC ENCODER ARCHITECTURE

The learned component of our model is an atomic encoder $f_\theta$ that converts noisy input maps into estimated atomic offsets. The architecture integrates a 3D volumetric feature extractor with a coordinate-based transformer to model protein ensembles. The volume encoder relies on a 3D ResNet-10 backbone to encode $64^3$ cryo-EM voxel grids into $N$-dimensional latent representations, where $N$ is equivalent to the number of atoms in the structure for convenience. These latent vectors are subsequently refined by a Diffusion Transformer (DiT), whose input consists of the latent vectors concatenated along the feature dimension with the coordinates of the reference structure. To accommodate variable-length sequences within a batch, proteins are zero-padded to the maximum

sequence length and handled via a binary attention mask to prevent padding tokens from influencing the self-attention mechanism.

The DiT employs adaptive layer normalization (AdaLN) to inject conditioning signals, such as the log-SNR of the current diffusion timestep, directly into the transformer layers via learned scale, shift, and gate parameters. To maintain the sequential order of the protein chain, the model applies a sinusoidal positional embedding to the input tokens, which is initialized with a standard deviation of 0.02. The atomic offsets output by $f_\theta$ are added to the reference coordinates $X_{\text{ref}}$ via a residual connection to produce the estimated coordinates of the input map.

## A.4 DATA SIMULATION

We retrieve six calmodulin proteins with similar structure from the PDB with the following IDs: 1EXR [Wilson & Brunger (2000)], 1OOJ [Symersky et al. (2003)], 1RFJ [Yun et al. (2004)], 1UP5 [Rupp et al. (1996)], 4CLN [Taylor et al. (1991)], and 5A2H [Kumar et al. (2016)]. The first four proteins are used for training, while the latter two are for validation. Structural motion is simulated using normal mode analysis along the first mode, which for all proteins corresponds to a twisting movement along the central alpha helix. The endpoints of this 4 Å motion are selected as the training structures, with the midpoint used to initialize the simulation serving as the static reference.

EMAN2 [Tang et al. (2007)] is used to generate density maps for each training structure at a grid size of $64^3$ with 2 Å per voxel and a resolution of 8 Å.

