# OpenReview forum: "Generative Modeling of Protein Conformational Ensembles With Cryo-EM Density Map Diffusion"
_ICLR.cc/2026/Workshop/LMRL — ICLR 2026 Workshop LMRL Poster_

### Official Review · Reviewer_mPeq · 2026-02-25
**Diffusion-Based Ensemble Modeling from Cryo-EM Density Maps and Static Reference Structures**

**Rating:** 7
**Confidence:** 3

**Review:**

### **Summary**

This paper introduces CryoSampler, a diffusion-based generative model for protein conformational ensembles that operates in cryo-EM density map space. Rather than diffusing directly in atomic coordinate space, the model performs diffusion in the space of density maps and learns to map noisy volumes to atomic coordinate offsets relative to a static reference structure. A differentiable cryo-EM forward model then renders predicted atomic coordinates back into density space, enabling supervision purely at the level of maps. Post-training, CryoSampler generates ensembles based purely on a reference structure with no density map required, unlike current inference-time methods CryoBoltz and ROCKET.

Importantly, the model does not require atomic models corresponding to each density map. Instead, it conditions on a static atomic reference structure and learns the distribution of atomic offsets that explain the observed ensemble of maps.

The method is evaluated on a synthetic dataset derived from six calmodulin proteins in the PDB. For each protein, normal mode analysis (ProDy) is used to simulate motion along the first mode, and cryo-EM maps are generated from these conformations using EMAN2. Four proteins are used for training and two for validation. The model successfully recovers bimodal conformational distributions for validation proteins and achieves near-angstrom RMSD to simulated states, outperforming AlphaFlow, Boltz-1, and a static reference baseline.

The core contribution - end-to-end diffusion on cryo-EM maps with an internal physics-based decoder - is conceptually strong and technically interesting. However, the evaluation is entirely synthetic and narrowly scoped, which limits the strength of the empirical claims.

### **Strengths and Weaknesses**

### **Strengths**

- Clear framing of ensemble prediction as a density-supervised problem.
- Elegant diffusion-based architectural design.
- The method does not require atomic labels for each density map, which is technically significant.
- Demonstrates recovery of bimodal conformational distributions in validation proteins.
- The inference pipeline is practical: only a static reference structure is required at test time.
- The distinction from inference-time methods (e.g., CryoBoltz, ROCKET) is clearly articulated.

### **Weaknesses:**
- The model is described throughout as being trained “only on cryo-EM density maps.” While it is true that supervision is applied solely in density space and no ground-truth atomic coordinates corresponding to each map are required, the method is fundamentally structure-conditioned. A static atomic reference structure is provided as input, and the model learns distributions of atomic offsets relative to this reference. Moreover, in the current experiments, the density maps themselves are generated from simulated conformational ensembles obtained via normal mode analysis. Thus, while the loss is density-supervised, structural inputs are essential to both training and inference. Clarifying this distinction would prevent confusion when the reference structure appears in Figure 1 and in the experimental setup.
- More explanation of normal mode analysis and EMAN2 would clarify the current experimental setup.
- All training data are synthetic: both conformations (normal mode analysis) and density maps (EMAN2) are simulated. While the authors are transparent about this, it weakens claims about leveraging cryo-EM data.
- The dataset is small (six closely related calmodulin proteins), limiting evidence of generalization beyond a single protein family.
- The model is stated to overfit training proteins to sub-angstrom accuracy (Table 1), which may indicate limited diversity in the simulated trajectories.
- Comparisons to AlphaFlow and Boltz-1 are somewhat unfair, as those methods are not trained on normal mode trajectories; the authors acknowledge this, but it still limits interpretability.
- It remains unclear how much conformational diversity is permitted under the structural loss constraints described in the appendix.
- The computational cost of generating simulated trajectories and maps with ProDy and EMAN2 is not reported.
- No experiments are performed on real heterogeneous cryo-EM datasets, which is ultimately the motivating application.
- When experiments are performed on real heterogeneous cryo-EM datasets, a benchmark against CryoBoltz and ROCKET would be appropriate. This benchmark could also be performed using EMAN2-derived density maps, as long as the potential handicap to CryoBoltz/ROCKET caused by synthetic data is explained.

### **Final Assessment:**
CryoSampler presents a technically elegant framework for ensemble modeling by performing diffusion directly in cryo-EM map space. The idea of implicit atomic model building within a generative model is compelling. However, the current validation relies entirely on synthetic data from a single protein family, limiting the empirical strength of the claims. Demonstration on real heterogeneous cryo-EM datasets would substantially strengthen the work.

---

### Official Review · Reviewer_9isc · 2026-02-25
**An interesting contribution that suffers from strong claims that lack empirical support.**

**Rating:** 6
**Confidence:** 3

**Review:**

The authors propose CryoSample, a diffusion model that generates conformational ensembles of proteins after training on ensembles of cryo-EM density maps and static reference structures. During inference, the model effectively predicts conformational ensembles from static reference structures.

$\textbf{Strenghts}$

The presented approach, to my best knowledge, is conceptually novel. The training approach, which centres on ensemble prediction as a map denoising problem, is interesting and warrants further investigation.

The paper addresses an open problem in a straightforward manner, and the preliminary results show that the approach works in principle.

$\textbf{Weaknesses}$

The benchmark is limited but, in my opinion, falls within what is expected for a workshop paper of this format. However, it can be argued that the comparison against AlphaFlow and Boltz-1 is misleading and that the paper would be stronger without it. However, the authors acknowledge that "As AlphaFlow and Boltz-1 are not trained on states generated with normal mode analysis, we should not expect them to perform well on our input data, but we report their accuracy for the sake of completeness". Instead, a comparison to existing methods presented in the introduction on the same data would have been interesting.

It would have been valuable to include a structural comparison of the training and validation structures to truly evaluate the generalisation ability. In addition, briefly evaluating the trained model on a less similar protein would have been interesting. This would not have taken away from the novelty but, especially for a workshop paper, may have added food for discussion regarding potential failure modes.

The scalability of the approach is not addressed, and model complexity as well as computational resource requirements are not reported. The model complexity would be important in the context of overfitting, as this is the expected behaviour of most architectures of sufficient capacity.
I furthermore believe that this information would be integral when reporting a new methodology/architecture at a workshop, to set expectations.

No source code is provided.

Overall, I believe it is an interesting contribution to a workshop that suffers from strong claims that are not (yet) empirically supported. I moved the rating from 7 to 6 due to a lack of source code.

---

### Meta-Review · Area_Chair_5HGt · 2026-02-27

**Recommendation:** Accept (Poster)
**Confidence:** 4

**Metareview:**

Accept.

---

### Decision · Program_Chairs · 2026-03-02

**Decision:**

Accept (Spotlight)

**Comment:**

Please see the meta-review.